# Early Diagnosis of Pseudohypoparathyroidism before the Development of Hypocalcemia in a Young Infant

**DOI:** 10.3390/children9050723

**Published:** 2022-05-15

**Authors:** Su Kyeong Hwang, Ye Jee Shim, Seung Hwan Oh, Kyung Mi Jang

**Affiliations:** 1Department of Pediatrics, School of Medicine, Kyungpook National University, Daegu 37224, Korea; skhwang@knu.ac.kr; 2Department of Pediatrics, Keimyung University School of Medicine, Keimyung University Dongsan Hospital, Daegu 37224, Korea; yejeeshim@dsmc.or.kr; 3Department of Laboratory Medicine, Pusan National University School of Medicine, Pusan National University Yangsan Hospital, Yangsan 50612, Korea; paracelsus@pusan.ac.kr; 4Department of Pediatrics, Yeungnam University School of Medicine, Yeungnam University Hospital, Daegu 42415, Korea

**Keywords:** pseudohypoparathyroidism, *GNAS*, albright’s hereditary osteodystrophy, osteoma cutis, hypothyroidism

## Abstract

Pseudohypoparathyroidism (PHP) is a rare, heterogeneous disorder characterized by end-organ resistance to parathyroid hormone (PTH). PTH resistance causes elevated PTH levels, hypocalcemia, and hyperphosphatemia. Since hypocalcemia causes life-threatening events, early diagnosis is crucial. However, the diagnosis of PHP is elusive during infancy because PHP is usually diagnosed with hypocalcemia-induced symptoms, which develop later in childhood when calcium requirements increase. A 1-month-old girl was referred to our clinic for elevated thyroid-stimulating hormone (TSH) levels on newborn screening. When measured 1 month after levothyroxine treatment, her TSH level normalized. At 4-months-old, multiple hard nodules were noted on her trunk. A punch skin biopsy revealed osteoma cutis associated with Albright’s hereditary osteodystrophy, a major characteristic of PHP. We performed targeted sanger sequencing of the *GNAS* gene and detected a heterozygous variant c.150dupA (p.Ser51Ilefs*3) in both the proband and her mother, causing frameshift and premature termination mutations. The patient was diagnosed with PHP Ia when she had normal calcium, phosphorous, and PTH levels. We report the early diagnosis of PHP Ia without hypocalcemia. It emphasizes the importance of meticulous physical examination in patients with congenital hypothyroidism.

## 1. Introduction

Pseudohypoparathyroidism (PHP) is a rare, heterogeneous disorder characterized by end-organ resistance to parathyroid hormone (PTH). It is caused by defects in the α-subunit of the stimulatory G protein (G_s_α), which is coupled to the PTH receptor [1,2]. Because of its rarity, its exact prevalence is unknown, and only two large studies exist on this condition. Its prevalence was estimated to be 0.34 in 100,000 and 1.1 in 100,000 according to reports from Japan [3] and Denmark [4], respectively. PHP is classified into types Ia, Ib, Ic, and II, which share biochemical features resulting from PTH resistance. PHP type I is further differentiated into types Ia, Ib, and Ic by the presence or absence of Albright’s hereditary osteodystrophy (AHO). The disorder is characterized by heterogeneous features (e.g., brachydactyly, a rounded face, central obesity, subcutaneous ossifications, short stature, and mental retardation), making it challenging to diagnose PHP through AHO [5,6,7].

PHP and hypoparathyroidism have similar treatments and clinical manifestations caused by hypocalcemia. However, PTH resistance causes elevated PTH levels, hypocalcemia, and hyperphosphatemia. These clinical symptoms of PHP are highly diverse, even when patients have the same genetic alterations. Many PHP patients experience hypocalcemia-induced life-threatening events such as syncope, seizure, paresthesia, and tetany [8,9]. Therefore, it is crucial to diagnose PHP early to avoid complications. However, PHP is difficult to diagnose during infancy because it is usually diagnosed with hypocalcemia-induced symptoms and hypocalcemia often develops later in childhood when calcium requirements increase.

Among the different types of PHP, type Ia involves multiple endocrine organs, such as the thyroid gland, ovaries, and pituitary gland, via G_s_α coupled receptors [10,11]. Hypothyroidism, a component of PHP type Ia, is often overlooked because it is mild and may be interpreted as congenital hypothyroidism, which is common in newborn screening [12]. There have been several attempts to diagnose PHP early and prevent complications; nevertheless, delayed diagnosis still occurs often [13,14]. Only a few cases of PHP have been diagnosed before the development of hypocalcemia and related symptoms [15,16,17,18]. Here, we report early diagnosis of PHP type Ia in a young infant before the development of hypocalcemia and elevated PTH levels owing to osteoma cutis. We emphasize the importance of meticulous physical examination in patients with congenital hypothyroidism.

## 2. Case Report

A 1-month-old girl was referred to our clinic for elevated thyroid-stimulating hormone (TSH) levels on newborn screening. She was delivered by cesarean section at full term, with a birth weight of 2.8 kg. Laboratory tests revealed elevated TSH (25 µU/mL, normal range 0.5–4.8) and normal free T4 (0.9 ng/dL, normal range 0.8–1.6) levels. The patient had no family history of thyroid disease, and the patient’s mother had no thyroid disease during the perinatal period. The patient had no goiter and thyroid antibodies. Thyroid ultrasonography showed unremarkable findings. Therefore, the patient was considered to have congenital hypothyroidism. Levothyroxine treatment was commenced (10 μg/kg/d), and a month later, her TSH level normalized. Although the patient was assumed to have congenital hypothyroidism, thyroid function tests were performed every 3 months. The patient was also carefully examined to rule out other causes of hypothyroidism. At 4 months old, multiple hard nodules were noted on her trunk with normal thyroid function. A punch skin biopsy revealed osteoma cutis associated with AHO, which is a major characteristic of PHP (Figure 1). The patient’s weight was 6.8 kg (50–75th percentile), and her height was 60.3 cm (15–25th percentile). There was no obvious obesity observed, which is an important indicator for early diagnosis of PHP. It was difficult to assess the patient’s exact developmental status owing to her young age. However, the patient seemed to follow normal developmental progress.

PHP type Ia was suspected owing to osteoma cutis combined with congenital hypothyroidism. Therefore, we performed targeted sanger sequencing of the *GNAS* gene (*NM_000516.5*), which is associated with PHP. We detected the heterozygous variant c.150dupA (p.Ser51Ilefs*3) in both the proband and her mother. It caused frameshift and premature termination mutations (Figure 2) and was classified as “pathogenic” according to the American College of Medical Genetics and Genomics/Association for Molecular Pathology guidelines, with evidence based on the following criteria: PVS1, PM2, and PP3. There is no reported allele frequency in the general population (Genome Aggregation Database) (=0%). Further laboratory tests revealed normocalcemia (9.7 mg/dL, normal range 8.8–10.8), normophosphatemia (5.7 mg/dL, normal range 3.8–6.5) and a normal serum PTH level (44 pg/mL, normal range 10–65).

*GNAS* is an imprinted gene. Maternally transmitted mutations cause PHP type Ia, while paternal transmission results in pseudopseudoparathyroidism (PPHP). Thus, the patient was diagnosed with PHP type Ia. Conversely, the patient’s mother was assumed to have PPHP; she had normocalcemia, normophosphatemia, normal thyroid function, and a normal serum PTH level.

The mother’s height and weight were 150 cm and 50 kg, respectively. She had short stature and a rounded face. Although these were clinical manifestations of AHO, they were mild and not accompanied by other AHO features, such as mental retardation, hard nodules, obesity, and brachydactyly. To avoid the onset of severe hypocalcemic symptoms, serum calcium, phosphorus, and PTH levels were measured every 3 months, and meticulous physical examination was done. At 1-year-old, the patient’s serum PTH level had markedly increased to 550 pg/mL, while calcium and phosphorous concentrations remained within normal ranges at 8.7 and 6.5 mg/dL, respectively.

Since elevated PTH precedes elevated serum phosphorous levels and hypocalcemia, regular laboratory tests were performed every 2 months. Two months later, elevated PTH (699 pg/mL) accompanied by hypocalcemia (6.9 mg/dL) and hyperphosphatemia (7.2 mg/dL) were noted. However, there were no hypocalcemia-induced symptoms. Calcitriol (30 ng/kg/d), and calcium carbonate (50 mg/elemental calcium/kg/d) were promptly administered. Both the hypocalcemia and elevated PTH level improved to 7.5 mg/dL and 580 pg/mL, respectively, 2 months later. The patient’s calcium level gradually improved as a result. Currently, she is 3 years and 7 months old, her height at the time was 97.3 cm (25–50th percentile) and her weight was 19.9 kg (99th percentile), and shows developmental milestones. However, she has not shown any hypocalcemia-induced symptoms.

## 3. Discussion

PHP is caused by de novo or autosomal dominant inheritance of inactivating mutations in the *GNAS* locus on 20q13.22 [14]. *GNAS* is an imprinted gene encoding G_s_α, which is coupled to the PTH receptor, thereby activating adenylyl cyclase. PHP is classified into PHP type I and PHP type II according to the response to bovine PTH. Among the types, PHP type Ia is caused by inactivating mutations in the maternal allele of the *GNAS* exons 1–13 [11,14,19,20]. Predominantly maternal *GNAS* expression has been detected in the thyroid gland, ovaries, proximal tubules of kidneys, and pituitary gland [2]. Therefore, patients with PHP type Ia also develop resistance to other hormones, such as TSH, growth hormone-releasing hormone, and gonadotropin, via Gs-coupled receptors.

AHO is associated with heterogeneous manifestations, including a round face, brachydactyly, short stature, and various degrees of mental retardation [21,22]. However, it is challenging to diagnose PHP early based on only AHO features because of its vague and nonspecific manifestations, which typically develop during late childhood [1,23]. In this case, the patient’s mother had only AHO without PTH resistance; she was assumed to have PPHP, as PPHP is caused by paternal inactivating mutations. 50% reduction of G_s_α caused by *GNAS* mutation leads to haploinsufficiency in tissues related the AHO phenotype such as the growth plate [24]. The variable expression of AHO with the same *GNAS* mutation suggest epigenetic modifications, other genetic loci, or environmental factors [22,25]. Moreover, recent studies related to obesity and cognition suggest that *GNAS* imprinting is also involved in other parts of the central nervous system [26,27]

There has been no report of PTH resistance at birth that develops over time from 0.1 to 22 years [20,28,29]. In general, PTH elevation is followed by hypocalcemia, and PHP type Ia is mainly diagnosed based on severe hypocalcemic symptoms such as seizures, paresthesia, and tetany, when calcium requirements increase during childhood. Therefore, it is difficult to identify at a young age. Moreover, its symptoms are often overlooked and treated as epilepsy, which further delays diagnosis.

There have been some attempts to diagnose PHP early using reported characteristics such as small for gestational age, early-onset obesity, cognitive impairment, developmental delay, and transient hypothyroidism [2]. However, such symptoms are difficult to notice in a very young infant, and cognitive development is normal in 30% of PHP type Ia cases. As such, there have only been five cases, including the present report, of genetically confirmed PHP Ia in young infancy before the development of hypocalcemia, providing detailed information [16,17,18,29] (Table 1).

Among the five cases of early diagnosis, four patients exhibited hypothyroidism. Although TSH resistance being reported as an early manifestation of PHP type Ia, this condition is often misdiagnosed as primary hypothyroidism and physical examination is skipped in patients with congenital hypothyroidism because of the rarity of PHP [30]. TSH resistance may be milder than PTH resistance due to partial G_s_α activity in the thyroid tissue [31]. There is a potential for higher prevalence of PHP type Ia among patients with elevated TSH and eutopic thyroid gland [32]. Therefore, meticulous physical examinations and checking calcium, phosphorous and PTH levels are needed especially in patients that are observed to have a mild increase in TSH concentration and presence of eutopic thyroid gland at first visit.

Osteoma cutis is one of the most common manifestations of AHO. Ectopic ossifications are caused by G_s_α deficiency in mesenchymal stem cells, resulting from de novo extraskeletal osteoblast formation in the dermis [2,33]. Consequently, osteoma cutis can develop in patients with PHP who have normal serum calcium, phosphate, and PTH levels [33,34]. Fortunately, in patients with PHP, osteoma cutis presents at birth or in early life, usually involving a single plaque or an isolated nodule [34]. Among the literature review cases, three cases showed AHO, however, ectopic ossification was pivotal in early diagnosing PHP type Ia during infancy in our patient and the literature cases.

Although PTH levels increase over time, elevated TSH may be present at birth. Therefore, patients are often diagnosed during newborn screening [2,29]. One large study of PHP showed that all of the young infant patients showed resistance to TSH [29]. Moreover, up to 30% of patients with PHP1A can be diagnosed at birth through elevated TSH newborn screening [32]. If the hypothyroidism was assumed to be primary hypothyroidism in this case and the nodules were overlooked, neuromuscular symptoms or even life-threatening hypocalcemic complications may have occurred. This report and literature review emphasize the significance of thorough physical examination in congenital hypothyroidism. If necessary, active biopsy for lesions, sequencing for *GNAS* and checking serial calcium and PTH levels are required in patients receiving treatment for congenital hypothyroidism in the hopes of reducing the instance of missing a PHP diagnosis.

## Figures and Tables

**Figure 1 children-09-00723-f001:**
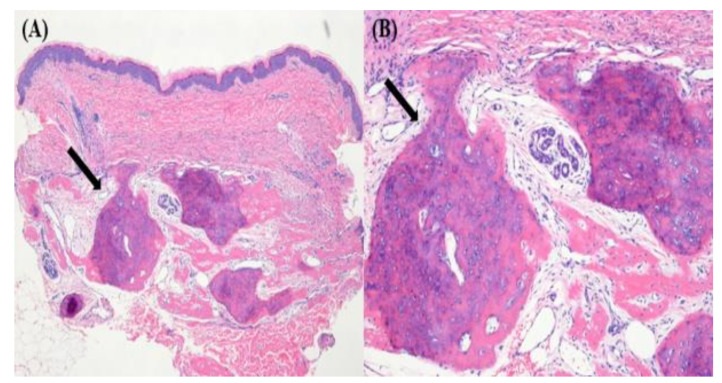
Histology slide showing a well-demarcated nodule, composed of mature lamellar bone within the dermis (black arrow). (**A**) 40× hematoxylin and eosin staining, (**B**) 100× hematoxylin and eosin staining.

**Figure 2 children-09-00723-f002:**
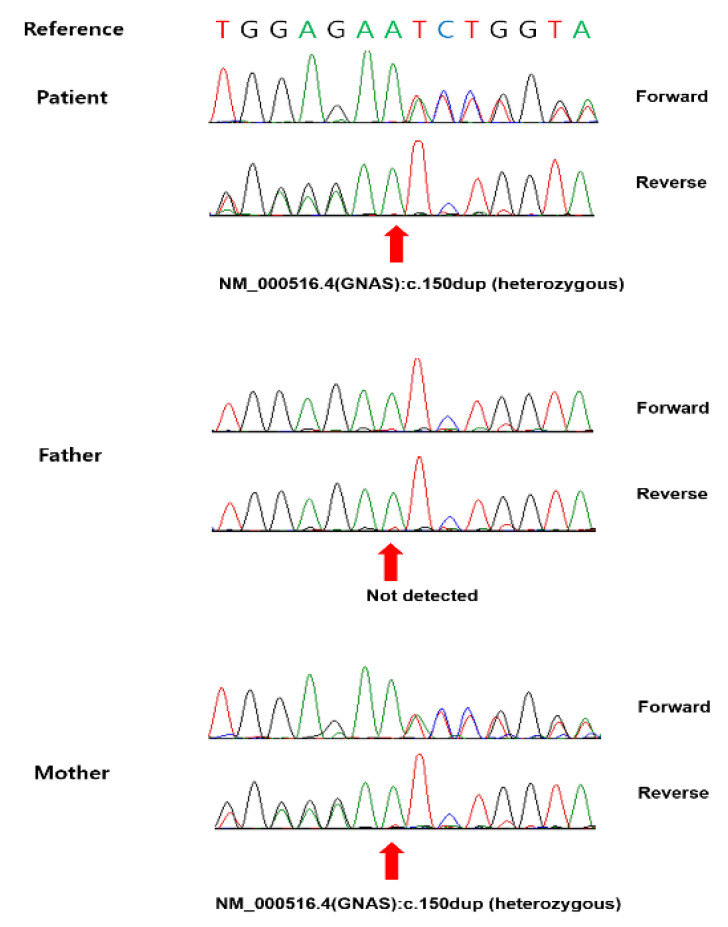
Genetic testing of the family. The patient harbored a c.150dupA (p.Ser51Ilefs*3) mutation (red arrow) inherited from her mother.

**Table 1 children-09-00723-t001:** Literature review of early diagnosed, genetically confirmed pseudohypoparathyroidism Ia before the development of hypocalcemia and related symptoms owing to osteoma cutis in young infants.

	Riepe, et al. [18]	Lubell, et al. [17]	Kodo, et al. [16]	Sano, et al. [29]	In Present Case
Age at diagnosis (months)	10	26	11	1	4
Sex	Male	Male	Male	NA	Female
Cause of referral	Hypothyroidism	Subcutaneous nodules	Developmental milestone	Subcutaneous nodules	Hypothyroidism
Site of osteoma cutis	Extremities and abdominal wall	Left thumb	Both of arms and legs, abdominal wall	NA	Abdominal wall
Family history	NA	Yes	Yes	NA	Yes
HGVS nucleotide (NM_000516.5)	c.317T>G	c.1100_1101insA	c.862dupA	NA	c.150dupA
HGVS amino acid	p.Ile106Ser	p.Asp368Glyfs*3	p.Ile288Asnfs*12	NA	p.Ser51Ilefs*3
Serum Calcium level (mg/dL)	8.98	9.3	10.0	10.42	9.7
Serum phosphorous level (mg/dL)	5.91	6.3	5.0	6.93	5.7
Serum PTH level (pg/mL)	202	237	42	93	44
Free T4 level before medication (ng/dL)	0.7	1.0	0.87	0.7	0.9
TSH level before medication (µIU/mL)	50.6	12.8	3.01	27.9	25

HGVS, Human Genome Variation Society; NA, not applicable; PTH, parathyroid hormone; TSH, thyroid-stimulating hormone. Calcium normal range: 8.8–10.8 mg/dL; Phosphorous normal range: 3.8–6.5 mg/dL; PTH normal range: 10–65 pg/mL; Free T4 normal range: 0.8–1.6 ng/dL; TSH normal range: 0.5–4.8 µIU/mL.

## Data Availability

The data presented in this study are available on request from the corresponding author. The data are not publicly available due to privacy.

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
