# Peer review of "Early Diagnosis of Pseudohypoparathyroidism before the Development of Hypocalcemia in a Young Infant"

_children, 2022, doi:10.3390/children9050723_

Round 1

Reviewer 1 Report

In the manuscript " Early diagnosis of pseudohypoparathyroidism before developing hypocalcemia in a young infant and literature review", the authors present a case report along with a literature review of pseudohypoparathyroidism. The paper could have a clinical significance and an impact in the clinical practice. However, minor criticisms are present, as follows:

- The title should be changed to Early diagnosis of pseudohypoparathyroidism before developing hypocalcemia in a young infant, remove the literature review from the title, as the literature studies are not so many and in the discussion part we have to discuss the literature results. Thus this paper is not a literature review;

- Lines 28-29 from the abstract should be removed;

- Lines 65-66 from the introduction is a repetition;

 - A conclusions should be added in order to better highlight the point of the case report.

Author Response

Thank you for giving me the opportunity to submit a revised draft of my manuscript titled “Early diagnosis of pseudohypoparathyroidism before the development of hypocalcemia in a young infant” to Children. 

We really appreciate the kind evaluation of our manuscript.

Reviewer 2 Report

Hwang et al. reported a girl diagnosed with PHP Ia when she had normal calcium, phosphorous, and PTH levels. The authors reviewed the literature regarding genetically confirmed PHP Ia before developing hypocalcemia in young infants, and described that the presented patient is the earliest diagnosed case of PHP before developing hypocalcemia.

AHO (obesity, round face, subcutaneous osteoma) in infancy can be a trigger to find it, but not many. Cases with increased TSH levels detected in neonatal mass screening have been  reported, yet few cases leading to the diagnosis of PHP. It is more common to find hypocalcemia after infancy (during febrile convulsions, etc.). In that sense, this is an important paper. However, the reviewer has some concerns.

Major comment 1
The authors described that the presented patient is the earliest diagnosed case of PHP before developing hypocalcemia. However, Sano et al. reported the patient diagnosed case of PHP before developing hypocalcemia at 1 month of age (J Endocr Soc. 2017 Nov 21;2(1):9-23. doi: 10.1210/js.2017-00293). Therefore, this patient is not the earliest diagnosed case of PHP before developing hypocalcemia. Furthermore, the authors need to cite this paper. 

Major comment 2
This patient with maternal GNAS mutation showed mild PHP phenotype and her mother with same mutation also had mild AHO features. The authors need to discuss about the mechanism of these mild phenotype.

Major comment 3
When she was referred to the authors for elevated TSH at one month of age, did you check her serum calcium level?

Author Response

Thank you for giving me the opportunity to submit a revised draft of my manuscript titled “Early diagnosis of pseudohypoparathyroidism before the development of hypocalcemia in a young infant” to Children. 

We really appreciate the kind evaluation of our manuscript. 

Please see the attchment

Round 2

Reviewer 2 Report

The authors responded appropriately to the reviewers' comments. We believe that this paper should be accepted.